# Insights into the Role of Natural Polysaccharide-Based Hydrogel Wound Dressings in Biomedical Applications

**DOI:** 10.3390/gels8100646

**Published:** 2022-10-12

**Authors:** Ying Sun, Duanxin Li, Yang Yu, Yongjie Zheng

**Affiliations:** 1College of Light Industry and Textile, Qiqihar University, Qiqihar 161000, China; 2Engineering Research Center for Hemp and Product in Cold Region of Ministry of Education, Qiqihar 161006, China

**Keywords:** natural polysaccharide, hydrogel, wound dressing, antibacterial and anti-inflammatory, application research

## Abstract

Acute skin damage caused by burns or cuts occurs frequently in people’s daily lives. Such wounds are difficult to heal normally and have persistent inflammation. Wound dressings not only improve the speed of wound healing, but also protect and cover the wound well. Hydrogels have the characteristics of good flexibility, high water content, and good biocompatibility, and are widely used in biomedicine and other fields. Common hydrogels are mainly natural hydrogels and synthetic hydrogels. Hydrogels cross-linked using different raw materials and different methods have different performance characteristics. Natural hydrogels prepared using polysaccharides are simple to obtain and have good biocompatibility, but are inferior to synthetic hydrogels in terms of mechanical properties and stability, and a single polysaccharide hydrogel cannot meet the component requirements for wound healing. Therefore, functional composite hydrogels with high mechanical properties, high biocompatibility, and high antibacterial properties are the current research hotspots. In this review, several common polysaccharides for hydrogel synthesis and the synthesis methods of polysaccharide hydrogels are introduced, and functional composite hydrogel dressings from recent years are classified. It is hoped that this can provide useful references for relevant research in this field.

## 1. Introduction

The skin can isolate germs, so that the human body is not harmed by germs. However, due to accidents such as burns and bumps, the human skin will be damaged [1]. Although the human body has a certain ability to repair itself, the wound healing cycle is long [2]. Without the barrier of the skin, the human body cannot isolate germs, and the germs invade the human body and cause wound infection. In addition, wounds produce large amounts of wound exudate during healing. In order to protect the wound from being infected by bacteria and absorb a large amount of wound exudate during wound healing, wound dressings came into being [3].

As a temporary skin substitute [4,5], wound dressings play an important role in wound healing. Traditional dressings, such as gauze, cotton pads, and bandages, can stop bleeding, absorb wound exudate, and protect the wound from bacterial infection [6,7], but hinder the wound healing process [8]. As a traditional dressing, gauze is still widely used in clinical practice. However, with research on wounds, it is found that this dry dressing very easily loses water and adheres to the wound site, which easily causes discomfort to patients when changing. The effect of gauze on preventing microbial invasion is poor [9]. Translucent film dressing is one of the commonly used wound dressings. This kind of film has the advantages of high transparency, easy operation, durability, and low price. At the same time, it has been widely studied and applied because of its good gas permeability and its ability to isolate bacterial pollution. Although film dressings can promote wound healing, due to their limited absorption capacity, they can only be used for shallow wounds containing a small amount of exudate, but not for highly exudate wounds [10]. Foam or sponge dressing can absorb exudate from deep wounds, and is suitable for deep wounds, diabetic ulcers, mild burns, and ulcers with venous insufficiency. However, foam dressings also require preparation of a specific thickness to absorb exudate. They depend on specific polymer materials, are not easy to remove from the wound, and have a single function, which limits their application in wound healing [11]. With in-depth research on wound healing, more requirements have been put forward for the function of wound dressings. An ideal wound dressing should provide the wound with a moist healing environment, good breathability, and mechanical protection [12]; it can absorb wound exudate, protect the wound from contamination and bacterial infection, reduce wound surface necrosis, and stimulate the growth of growth factors [13], as well as good biocompatibility, biodegradability, and easy and painless removal and replacement [14].

Hydrogel is a water-soluble hydrophilic polymer material which can form a three-dimensional network structure function through physical or chemical cross-linking, and has the advantages of a high affinity for water and good biocompatibility [15]. As an emerging functional wound dressing, hydrogel dressings can keep the wound moist and absorb wound exudates continuously [16]. More importantly, through structural design and functional integration, hydrogel dressings can be endowed with a variety of excellent properties, and then play an important role in various processes of wound healing [17,18,19,20,21]. Compared with synthetic polymer hydrogels, hydrogel materials based on natural polymers have better biocompatibility and biodegradability [22]. Natural polysaccharides are biological macromolecules that widely exist in animals, plants, and microorganisms, and have good biological activity. A large number of basic research studies and clinical applications have confirmed the safety and effectiveness of polysaccharide-based hydrogel dressings in wound healing, which can accelerate wound healing, reduce scars, and relieve patients’ pain [23]. Some hydrogels prepared from natural polysaccharides can be degraded in time to avoid secondary damage when changing dressings [24]. At present, the research on polysaccharide-based hydrogel in the field of drug release is mostly aimed at a certain drug. A disease often causes a variety of complications during its development. The loading of two or more drugs and their dynamic controlled release can improve the effectiveness of drugs and increase the probability of curing diseases. It is a big challenge to prepare intelligent hydrogels with precise control, sequential release, and response to changes in external environment [25]. Compared with the traditional simple hydrogel dressing, the composite hydrogel dressing has better performance and effect in protecting the wound surface and promoting wound healing. Based on the above information, this review aims to recent advances in natural polysaccharide-based hydrogel wound dressings. Hyphenate -proliferation and formation of polysaccharide-based hydrogel wound dressing in a series of complex processes from wound formation to wound healing (hemostasis, antibacterial, anti-inflammatory, and tissue regeneration) (Figure 1).

## 2. Materials

Natural polysaccharide polymers have a wide range of raw materials, and hydrogels based on natural polysaccharides have the characteristics of high water retention, reproducibility, biodegradability, biocompatibility, and non-toxicity, making them suitable as medical materials [27,28,29,30,31]. Common natural polysaccharides used for hydrogel synthesis include glucan, chitosan, hyaluronic acid, alginic acid, and cellulose. The chemical structure formulas of common polysaccharides are shown in Figure 2.

### 2.1. Dextran

Dextran refers to a homopolysaccharide composed of glucose as a monosaccharide. There are a large number of free hydroxyl groups in the side chain, which is easy to modify and cross-link, and can be chemically modified by various methods such as sulfation, acetylation, carboxymethylation, or oxidative ring opening [32,33]. Dextran hydrogels can be obtained by introducing polymerizable groups into dextran for cross-linking [34,35]. Dextran has good biocompatibility and resistance to protein adsorption [36]. Its immunomodulatory function is considered to be the most important physiologically active function. It can activate macrophages and neutrophils, induce cytokine production, and effectively strengthen the immune system. It has attracted much attention in the treatment of infectious diseases and trauma [37]. Therefore, dextran is widely used in fields such as biopharmaceuticals and biotechnology.

### 2.2. Chitosan

Chitosan is a natural alkaline polysaccharide, generally prepared from chitin by deacetylation. Chitin is widely found in algae, mollusk shells, and animal bones, and is widely found in nature [38]. Chitosan can be mixed with negatively charged macromolecules (such as chondroitin sulfate, sodium alginate, etc.) to form gels, and hydrogels can be formed by graft modification of chitosan hydroxyl groups [39,40]. Chitosan has good biocompatibility and can be enzymatically decomposed into harmless products such as low-molecular-weight chitosan oligosaccharide and glucosamine, which will not cause immune rejection in human tissues [41,42]. 

### 2.3. Hyaluronic Acid

Hyaluronic acid is a non-sulfated anionic, hydrophilic glycosaminoglycan produced by streptococcus for industrial use, and it is also present in human connective tissue and providing lubrication plays an important role [43]. Hyaluronic acid is an attractive biomaterial due to its unique viscoelasticity and immunogenicity [44], which is widely used in the treatment of joint diseases. As a novel biocompatible and biodegradable material, hyaluronic acid hydrogels have been widely used in tissue engineering and drug delivery [45].

### 2.4. Alginic Acid

Alginic acid is a natural polysaccharide present in the cell walls of brown algae, and the pure product is white-to-brownish-yellow fibers, granules, or powders. Alginic acid easily forms gels with cations, such as the cross-linking of sodium alginate and Ca^2+^ [46]. Due to the large molecular weight of alginate, it is difficult to degrade, and the lack of binding sites for reacting with amino acids, proteins, and other substances limits the application of alginate as an in vivo graft to a certain extent [47]. In order to further enhance the practicality of alginate, researchers modified the structure of alginate and introduced more active groups or side chain molecules [48], which successfully expanded the application of alginate in the field of tissue engineering. Due to its hydrophilicity, alginate can absorb a large amount of wound exudate, and shows hemostatic effect when it comes into contact with bleeding wounds [49]; sodium alginate can also promote the formation of blood vessels and the production of collagen by promoting cell migration, reduce the concentration of pro-inflammatory cytokines in chronic wounds, and promote wound healing [50]. 

### 2.5. Cellulose

Cellulose is one of the most common macromolecular compounds and is a renewable resource. It mainly exists in plants, but can also be produced by certain bacteria (such as Acetobacter, Agrobacterium, Pseudomonas, etc.), algae, and fungi [51,52]. Cellulose molecules are rich in hydroxyl groups, which can be used to prepare hydrogels with different structures and properties [53]. Cellulose hydrogels can be prepared from natural cellulose (including bacterial and plant sources) or cellulose derivatives (such as methylcellulose, carboxymethylcellulose, etc.) [54]. Cellulose is an attractive precursor material for hydrogels, which is widely available, inexpensive to regenerate, and has good biocompatibility [55,56,57]. Some cellulose hydrogels also respond to changes in external stimuli. Based on the above advantages, cellulose-based hydrogels can be used in many fields [58,59], and their scaffold materials are widely used in the regeneration of various tissues, such as bone, cartilage, heart, blood vessels, nerves, and liver [60].

## 3. Methods

According to the preparation method, polysaccharide hydrogel materials can be divided into physical cross-linked hydrogels, chemical cross-linked hydrogels, and physical/chemical hybrid double-cross-linked hydrogels, as shown in Figure 3 [61].

### 3.1. Chemical Cross-Linked Hydrogels

Chemically cross-linked hydrogels are the formation of new covalent bonds between polymer molecular chains. The interactions between the chains are tighter, and the resulting hydrogels generally have better structural stability and mechanical properties. Chemical cross-linking usually requires the introduction of crosslinking agents [62,63]. The main chain of polysaccharide molecules has a large number of hydrophilic groups (such as -OH, -COOH, and -NH_2_) [64]. By grafting functional groups suitable for crosslinking on polysaccharide molecules, modification of polysaccharide molecules can be achieved [65,66,67]. Chan chemically modified alginate with glutaraldehyde by acid-catalyzed acetalization to prepare hydrogels which are stimuli-responsive and exhibit high water absorption. It can be used as a drug delivery vehicle for protein therapy [68]. However, some organic crosslinkers (such as glutaraldehyde, epichlorohydrin, borax, acetic anhydride/oxalic acid, etc.) should be adequately removed after use. Residual crosslinking agents in large doses can cause cytotoxicity in hydrogels, and avoids the use of organic crosslinking agents. To solve this problem, Kanafi used citric acid as a natural cross-linking agent to construct pH-responsive carboxymethyl cellulose/polyoxyethylene hydrogels [69]. The use of natural crosslinking agents can avoid traditional crosslinking cytotoxicity of the agent. Another chemical crosslinking method is the formation of hydrogels by enzymatic catalysis. Hou used horseradish peroxidase (HRP) and H_2_O_2_ to act, and enzymatic crosslinking of polymer-phenol conjugates enabled the in situ synthesis of injectable hydrogel [70].

### 3.2. Physically Cross-Linked Hydrogels

Physically cross-linked hydrogels based on non-covalent interactions are network knots linked together by molecular chain entanglement or intermolecular interactions (such as hydrogen bonds, hydrophobic interactions, electrostatic interactions, and coordination bonds, etc.) [71,72]. No new chemical bonds are formed during the preparation. Polysaccharide hydrogels prepared by physical cross-linking can be formed under mild conditions, but usually have poor mechanical properties, poor long-term stability, and easy degradation [73]. On the other hand, the physical cross-linking method does not introduce chemical cross-linking agents, has low cytotoxicity, and has a positive effect on unstable biologically active substances and living cells.

### 3.3. Double-Cross-Linked Hydrogel

Natural polysaccharide-based double-crosslinked hydrogels include physical–chemical double-crosslinked hydrogels and physical–physical double-crosslinked hydrogels. Both physical and chemical crosslinking exist in the three-dimensional network of physical–chemical double-crosslinking hydrogels. Through the ion-covalent double-cross-linking mechanism, it contains two interpenetrating networks. One is rigid cross-linking through covalent bonds. The other is a loose physical network formed by cross-linking through molecular interactions (such as hydrogen bonds, ionic interactions, coordination interactions, etc.). The physical–physical double-crosslinking hydrogel maintains the hydrogel network through physical crosslinking in two different ways, which improves the gel strength while avoiding the hidden cytotoxicity of chemical crosslinking. Compared with the single-network physically cross-linked hydrogel and chemically cross-linked hydrogel systems, the double-cross-linked hydrogel network has higher mechanical strength and toughness [74,75]. These methods have solved the problem of poor mechanical performance of a single polysaccharide, making polysaccharide gel more suitable for specific fields such as wound dressings. Fan prepared composite water by oxidizing the Schiff base reaction between the aldehyde of konjac glucomannan (OKGM) and the amino group of carboxymethyl chitosan (CMCS), and adding different amounts of graphene oxide (GO) as a nanoadditive gel [76]. The schematic diagram of the hydrogel synthesis route is shown in Figure 4.

## 4. Results and Discussion

### 4.1. Functional Composite Hydrogel Wound Dressing

With the continuous deepening of research on hydrogels, hydrogels formed by chemical or physical crosslinking can no longer meet the needs of applications, and the construction of hydrogels has specific responsiveness to external stimuli (stress, temperature, pH, magnetic field, etc.) or they have special functions (self-healing, adhesion, electrical conductivity, frost resistance, etc.) [77,78,79]. Functional composite hydrogels have become the focus of research. Functional composite hydrogel dressings are based on hydrogels by introducing new components to cause them to have special functions or enhance their healing effects [80,81]. At present, there have been many studies on functional composite hydrogel dressings, such as sensor-containing hydrogel dressings and drug-loaded hydrogel dressings. 

#### 4.1.1. Sensor-Containing Hydrogel Dressings

pH is an important factor affecting wound healing, and the pH of wounds can be significantly increased under the induction of tissue damage and immune defense [82]. Under normal circumstances, the pH of a healing wound is in the range of 5 to 6, which is weakly acidic [83]. If the pH rises to slightly alkaline, it means that the wound may be infected [84,85]. According to the different dissociative groups, pH-sensitive hydrogels can be divided into anionic, cationic, and zwitterionic. In general, the volume of these three hydrogels changes with the change in pH value (Figure 5a). There is a smart antibacterial biomaterial based on keratin hydrogels with pH-dependent behavior and ZnO nanoplates as fungicides. The pH of the wound during bacterial metabolism is alkaline. Initially shrinking at acidic pH, the hydrogel swells upon contact with bacteria-contaminated media, resulting in the release of nanoparticles (Figure 5b).

The pH hydrogel dressing can monitor wound infection without removing the dressing, avoiding the invasion of external bacteria and reducing patient suffering [88]. Generally, the pH of the wound can be determined by implanting mesoporous microparticles or color-changing textiles containing pH dyes into the dressing, and only by observing the change in the color of the dressing [87,89,90,91]. As a new type of nanomaterial, carbon dots (CDs) have unique optical, electrochemical, biocompatibility, and photoluminescence properties, and can be used as pH monitoring sensors in hydrogels. Omidi proposed a novel chitosan/CD nanocomposite as an antibacterial wound healing bandage and conducted in vivo experiments on three groups of rats. The results showed that the chitosan/CD nanocomposite not only possessed high pH sensitivity, but also improved the wound healing process due to their antibacterial properties (Figure 6) [92].

In the process of wound healing, wound dressings play a great role in protecting and repairing damaged skin. Frequent replacement of dressings on wounds is not only unnecessary, but also greatly affects the recovery of wounds. In general, hydrogel dressings can go without changing for a long time. However, in some cases, the dressing has to be removed, such as wound infection due to improper wound management. Moreover, because these changes occur under the dressing, it is difficult to observe such subtle changes in time with the naked eye if the dressing is opaque. Therefore, compared with traditional dressings, the use of sensor-containing hydrogel dressings can judge the changes in wounds, detect possible infection problems of wounds as early as possible, effectively avoid large-scale bacterial infection of wounds, and reduce unnecessary dressing changes. 

#### 4.1.2. Drug-Loaded Hydrogel Dressings

Hydrogel dressings using polysaccharides or polysaccharides mixed with other polymer materials have been studied for a long time, and a large number of studies have confirmed the role of hydrogels in promoting wound healing. However, only relying on the healing-promoting properties of polysaccharide hydrogels still has certain limitations and cannot meet the diversified needs in the wound healing process. Therefore, in order to improve the effect of the hydrogel, some active substances beneficial to wound healing can be loaded on the hydrogel to enhance the wound healing effect. Table 1 lists the components of hydrogel composites and their functions in recent years.

##### Simple Delivery of Functional Hydrogel Dressings

The extracellular matrix is an intricate 3D network structure, and polysaccharide macromolecules are an important part. The extracellular matrix is a hydrophilic matrix. The polysaccharide-based natural hydrogel maintains its three-dimensional structure without dissolving, which makes the natural hydrogel structurally similar to the extracellular matrix with very high water content, good biocompatibility, and low immunogenicity, constructing a microscopic structure that resembles the natural extracellular matrix, thereby providing structural support and protection, as well as the ability of cells to interact in a three-dimensional environment, supporting cell growth and repair. The structure of the hydrogel is similar to the extracellular matrix, and the hydrogel facilitates the migration of cells and growth factors during wound healing and can also be used to deliver drugs. Based on the hydrogel matrix, ingredients beneficial to wound healing can be added to increase the healing-promoting effect. The simplest method is to directly add the substances used to the matrix solution before the synthesis of the hydrogel, and make it uniformly dispersed in the hydrogel colloid during the preparation process. Raghavendra prepared antibacterial chitosan–silver nanocomposite films (CSNFs) by microwave irradiation, and the results showed that the films had good antibacterial properties against Escherichia coli (*E. coli*) and Staphylococcus aureus (*S. aureus*) (Figure 7) [107].

In addition, the carrier can also be combined with the target substance by methods such as microsphere embedding, mesoporous particle filling, compound complexation, etc., and then the carrier is dispersed in the hydrogel [108,109]. These methods can prevent the inactivation of active substances in the early stage and can achieve the effect of slow release and sustained release.

##### Stimuli-Responsive Hydrogel Dressings

When the external conditions (temperature, pH, or light, etc.) change, the volume or structure of the hydrogel changes, which causes the release of the loaded substances in the hydrogel, which is called stimuli-responsive hydrogel dressing.

Temperature-responsive hydrogels refer to changes in the swelling degree and light transmittance of the hydrogels when the ambient temperature changes. As the stimuli response to temperature can occur under natural conditions, it can also be artificially adjusted to temperature changes. When the temperature changes, it will affect the hydrophobic interaction and hydrogen bonding in the molecular chain, thus affecting the structure of the hydrogel [110]. Adding poly(lactic acid-co-glycolic acid)-poly(ethylene glycol)-poly(lactic acid-co-glycolic acid) (PLGA-PEG-PLGA) triblock copolymer thermogel to the hydrogel, the use of elevated temperature will cause dehydration of the PEG chains, leading to contraction of the micelle formation, thereby enhancing the interaction between the PEG and PLGA blocks. At temperatures below the critical gelling temperature (CGT), bridging between micelles is usually unstable due to the low hydrophobicity of PLGA. As the temperature increases, the hydrogel is temperature-responsive due to the increase in the hydrophobic interaction between the chains, which enhances the intermicelle interaction and aggregation, leading to the subsequent gelation, as shown in Figure 8a. At lowering of the copolymer solution below its transition temperature, the copolymer solution was transparent. Upon heating, the copolymer solution became opaque without gelling until a gel was formed, which was not fluid upon inversion (Figure 8b) [111].

pH-sensitive hydrogels can respond to changes in environmental pH through changes in their shape or volume and release loaded substances [112]. Qu developed a series of polysaccharide-based self-healing hydrogels with pH-sensitivity as drug delivery vehicles for hepatocellular carcinoma therapy. The hydrogels were prepared by using N-carboxyethyl chitosan (CEC) and dibenzaldehyde-terminated polyethylene glycol (PEGDA) synthesized in aqueous solution. Doxorubicin (Dox), as a water-soluble small molecule anticancer drug, is encapsulated in hydrogels and released when pH changes (Figure 9) [113].

Light-responsive hydrogels can undergo processes such as network assembly–disassembly and sol–gel conversion when they are changed by external light. The intelligent response to light is stimulated by the introduction of light-responsive functional groups and remotely controlled in situ changes. Cho synthesized a new polysaccharide-based thermogel polymer that can be photo-crosslinked by UV irradiation to form hydrogels with mechanical elasticity. Methacrylated hexanoyl glycol chitosan (M-HGC) was synthesized by a series of chemical modifications of glycol chitosan (GC), N-hexanoylation, and N-methacrylate. Cell viability and cell encapsulation experiments showed good cell viability and chondrocytes continued to survive within the observed photo-crosslinked hydrogel matrix for 7 days (Figure 10). M-HGC polymers can be considered as promising biomaterials for various biomedical applications [110].

## 5. Conclusions

As a new type of medical dressing, hydrogel is a supplement and enhancement to the functions of other types of medical dressings, so that the medical dressing not only has good high elasticity, permeability, and strong water absorption, but also has good biocompatibility and compatibility. Adjusted physicochemical properties make it widely used in bacterial infections and difficult-to-heal wounds, it can overcome the limitations of high and low temperature environments, and it can be used normally. Compared with synthetic polymer hydrogels, hydrogel materials based on natural polymers have better biocompatibility and biodegradability. Natural polysaccharides are biological macromolecules that widely exist in animals, plants, and microorganisms, and have good biological activity. A large number of basic studies have confirmed the safety and effectiveness of polysaccharide-based hydrogel dressings in wound healing. They can speed up wound healing and relieve the pain of patients. They are relatively ideal wound dressings.

At present, improving the mechanical strength and healing-promoting ability of hydrogel materials is the main goal of developing hydrogel materials, mainly through the use of new natural polysaccharides, modification of natural polysaccharides, and blending of natural polysaccharides with other substances, beneficial to wound loads of healing active substances (such as various antibacterial drugs, cells, and stem cells, various growth factors, etc.). However, due to various constraints such as cost, safety, and the production process, some new composite hydrogel dressings are still in the laboratory research stage and have not been widely used for the time being. In the future, research on hydrogels and their dressings needs to be deepened, and improving their healing-promoting effect and mechanical properties is still the top priority of future research. This review introduces several common polysaccharides used for hydrogel synthesis and the synthesis methods of polysaccharide hydrogels, and summarizes the functional composite hydrogel dressings in recent years. Although some polysaccharide hydrogel dressings containing bioactive substances (such as growth factors and fibroblasts) have been commercialized, the application cost is high and there are certain safety risks. The relevant animal experiments and clinical applications are summarized. It is hoped that more bioactive hydrogel composites can be prepared through this research. It is possible to develop hydrogel dressing products with lower cost, better performance, and wider application range.

## Figures and Tables

**Figure 1 gels-08-00646-f001:**
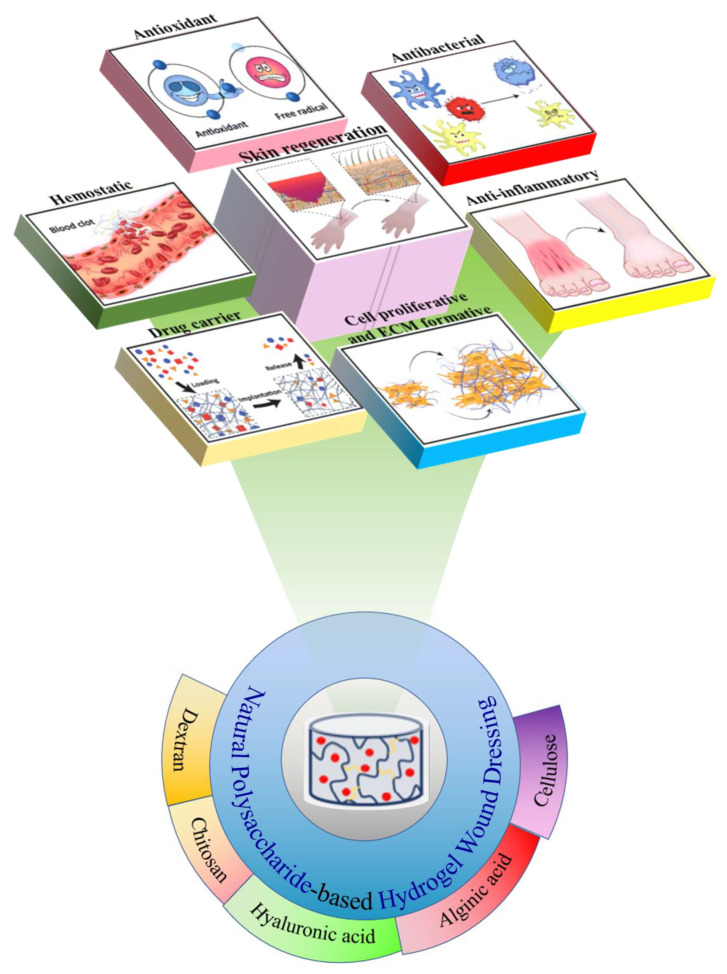
Hyphenate -proliferation and formation of polysaccharide-based hydrogel wound dressing. Reprinted with permission from Ref. [26]. Copyright 2021, copyright with permission from Ahmadian and Correia.

**Figure 2 gels-08-00646-f002:**
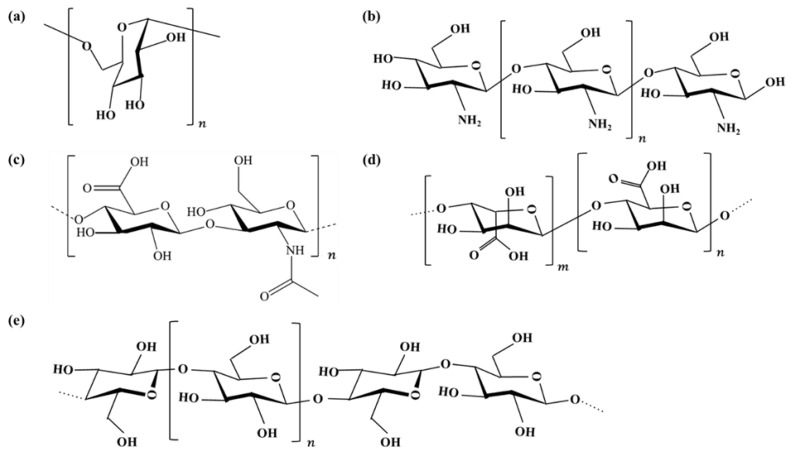
Chemical structural formulas of common polysaccharides: (**a**) Dextran; (**b**) Chitosan; (**c**) Hyaluronic acid; (**d**) Alginic acid; (**e**) Cellulose.

**Figure 3 gels-08-00646-f003:**
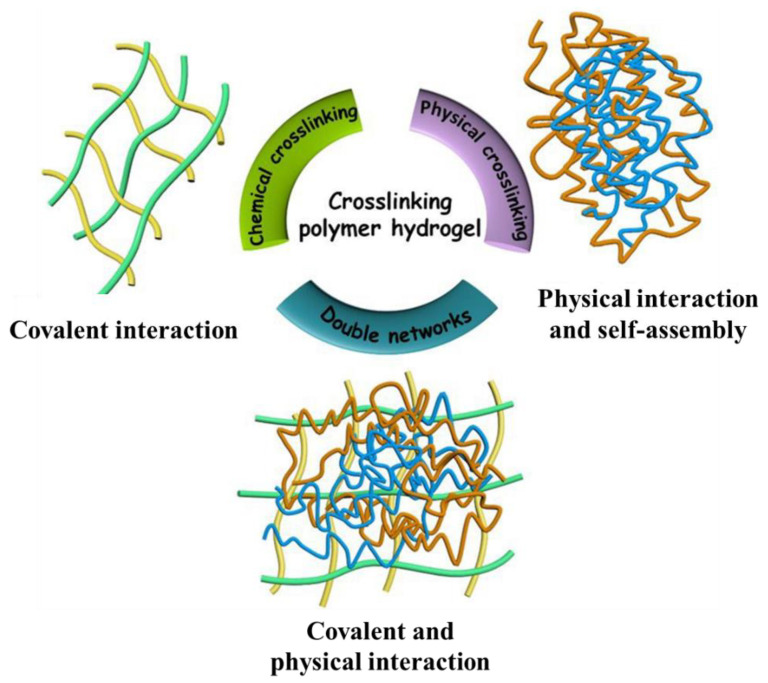
Schematic presentation of the hydrogels fabricated through chemical crosslinking, physical crosslinking, or double network. Reprinted with permission from Ref. [61]. Copyright 2020, copyright with permission from Nandi, Mondal and Das.

**Figure 4 gels-08-00646-f004:**
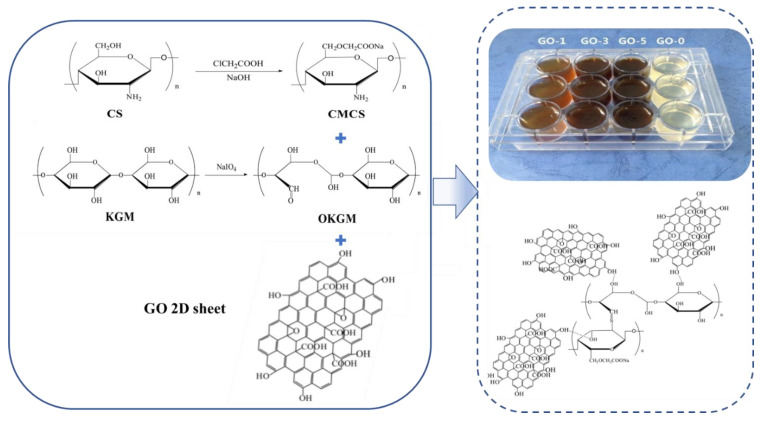
Schematic illustration of the synthesis route of the hydrogel. Reprinted with permission from Ref. [76]. Copyright 2016, copyright with permission from Fan and Yi.

**Figure 5 gels-08-00646-f005:**
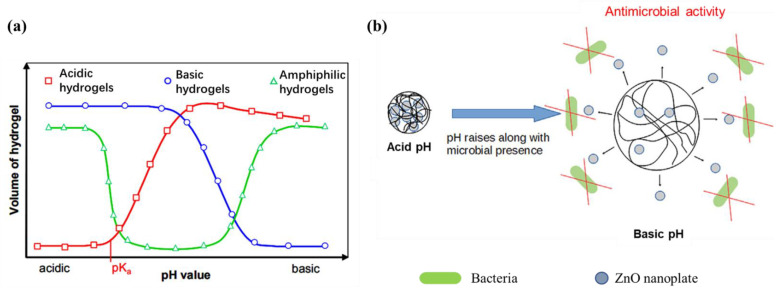
(**a**) pH-sensitive hydrogels volume change with the pH of solution; (**b**) The pH of the wound during bacterial metabolism is alkaline and the hydrogel swells to release ZnO nanoplate [86,87]. Reprinted with permission from Ref. [86]. Copyright 2008, copyright with permission from Andreas and Georgi. Reprinted with permission from Ref. [87]. Copyright 2018, copyright with permission from Villanueva and Cuestas.

**Figure 6 gels-08-00646-f006:**
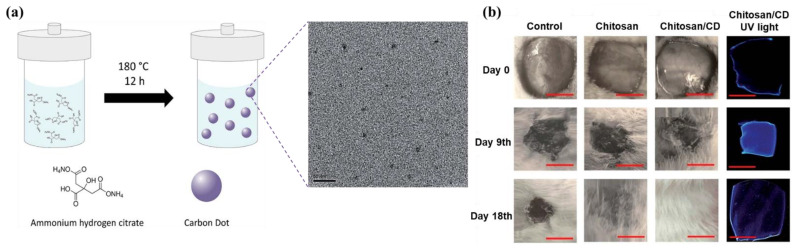
(**a**) Schematic illustration for synthesis of carbon dots and TEM images of CD; (**b**) The visual images of the untreated (control) and treated wound with chitosan, and chitosan/CD nanocomposite at different stages (0, 9, and 18 days); the fluorescence images at the right-hand side show the pH sensitivity of the chitosan/CD nanocomposite. Reprinted with permission from Ref. [92]. Copyright 2017, copyright with permission from Omidi and Yadegari.

**Figure 7 gels-08-00646-f007:**
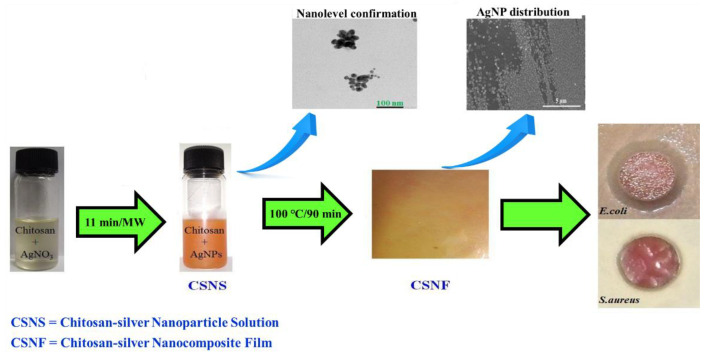
Pictorial illustration of green fabrication of chitosan–silver nanocomposite films (CSNFs) and their antibacterial efficiency. Reprinted with permission from Ref. [107]. Copyright 2016, copyright with permission from Jung and Raghavendra.

**Figure 8 gels-08-00646-f008:**
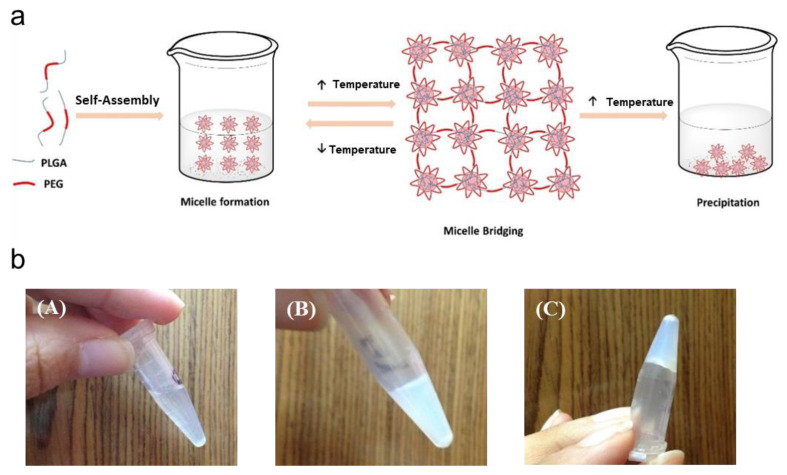
(**a**) Diagram of the sol–gel transition of BAB-type poly(lactic acid-co-glycolic acid)-poly(ethylene glycol)-poly(lactic acid-co-glycolic acid) (PLGA-PEG-PLGA) tri-block copolymeraqueous solution in response to temperature; (**b**) (**A**,**B**) are the photographs of a PLGA-PEG-PLGA copolymer solution prior to sol–gel transition (**A**) and formation of opaque solution (**B**), and thermogels produced after sol–gel transition retain their position (**C**). Reprinted with permission from Ref. [111]. Copyright 2018, copyright with permission from El-Zaafarany and Soliman.

**Figure 9 gels-08-00646-f009:**
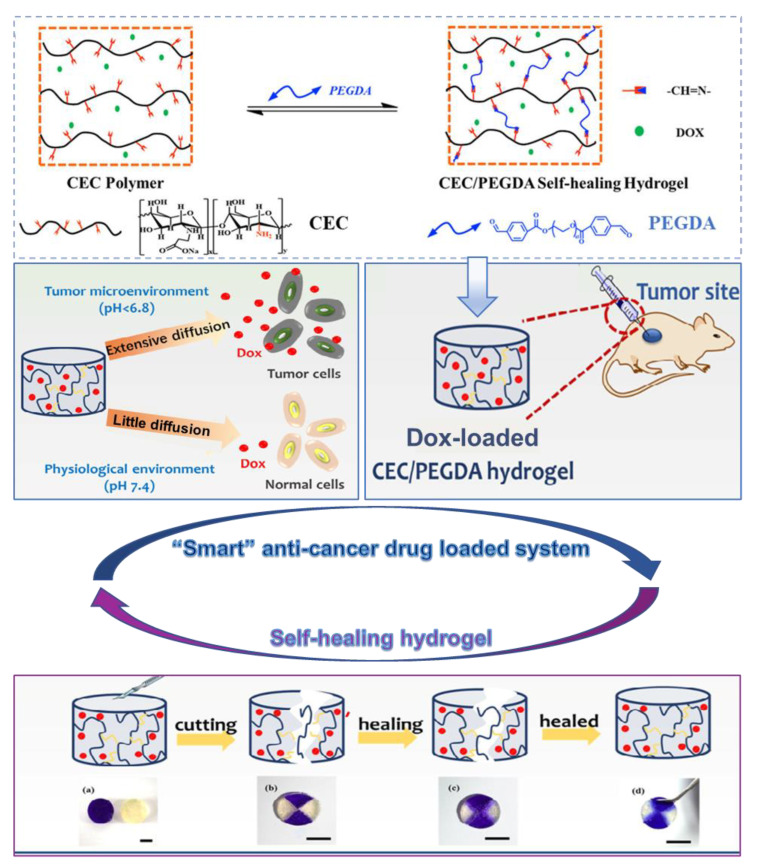
Self-healing hydrogels with pH-sensitivity as drug delivery vehicles for hepatocellular carcinoma therapy and photographs of self-healing performance of CEC/PEGDA hydrogels. (**a**) Two disk-shaped hydrogels (one stained with crystal violet and the other stained with FITC); (**b**) Hydrogels were cut into four equal pieces and put together; (**c**,**d**) The hydrogels healed completely into one block after 3 h at 25 °C. Reprinted with permission from Ref. [113]. Copyright 2017, copyright with permission from Qu and Zhao.

**Figure 10 gels-08-00646-f010:**
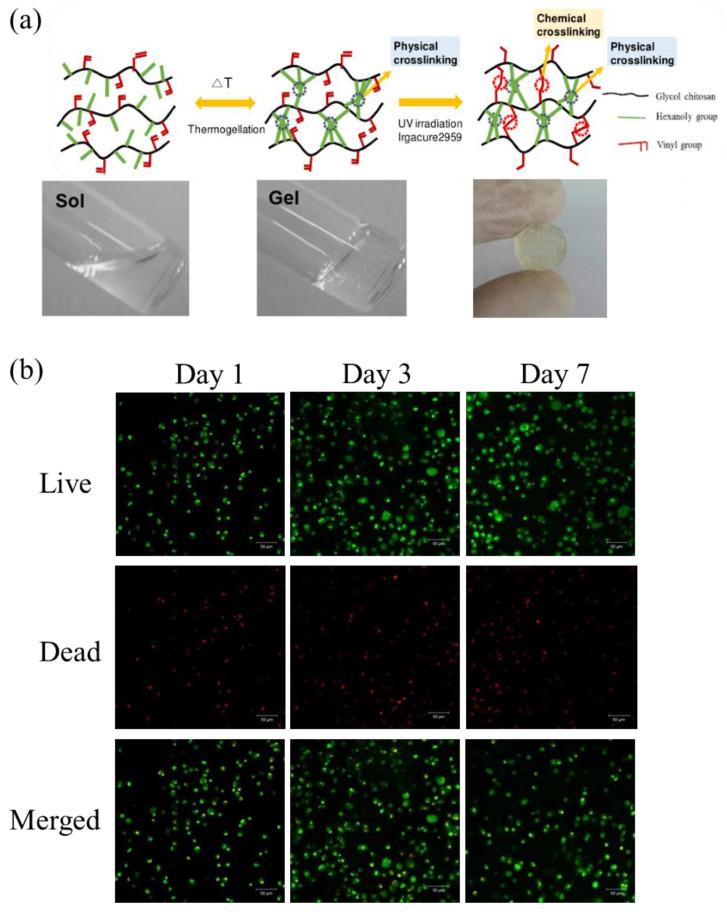
(**a**) Schematic illustration of thermo-reversible sol–gel transition and photo-345 crosslinked hydrogel formation upon UV irradiation; (**b**) Live/dead imaging of encapsulated chondrocyte cells within the photo-crosslinked M5-HGC hydrogels at 1, 3, and 7 days. Reprinted with permission from Ref. [110]. Copyright 2016, copyright with permission from Cho.

**Table 1 gels-08-00646-t001:** Loading substances of hydrogels and their functions.

Loading Substances	The Function of Loading Substances	Target Diseases	References
Metal ions: Ag^+^, Cu^+^, and Zn^+^, etc.	Inhibit bacterial metabolism, hinder bacterial DNA replication, and accelerate cell healing	Suitable for all kinds of skin injuries, especially acute and chronic wounds (bruises, dermal area, surgical incision, first- and second-degree burns, vascular or metabolic disease, treatment of ulcers, bedsores)	[93,94,95,96,97]
Drug Loading: Gentamicin and ampicillin, etc.	Antibacterial and prevent infection	Remove superficial wounds, progressively deteriorating wounds, and necrotic tissue in deep wounds, it can also be used in the treatment of granulated wounds, skin peels, and radiation burns, as well as in the treatment of infected wound	[98,99,100]
Growth factor: Vascular endothelial growth factor (VEGF), basic fibroblast growth factor (bFGF), and platelet-derived growth factor-BB (PDGF-BB), etc.	Prolonged cell proliferation activity and promote angiogenesis	Moisturizes wounds and is suitable for partial and full thickness wounds such as pressure ulcers, lower extremity ulcers, and diabetic ulcers	[101,102,103,104]
Cell: Human mesenchymal stem cells and human dermal fibroblasts	Promote wound healing	Involved in wound healing through cell recruitment and cytokines, promoting granulation, angiogenesis, epithelialization, collagen production	[105,106]

## Data Availability

The figures used and analyzed during the current review are available from the publisher and corresponding author on reasonable request.

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
