# Peer review of "Insights into the Role of Natural Polysaccharide-Based Hydrogel Wound Dressings in Biomedical Applications"

_gels, 2022, doi:10.3390/gels8100646_

Round 1

Reviewer 1 Report

The review article entitled "Insights into the role of natural polysaccharide-based hydrogel wound dressings in biomedical applications" systematically presented an update on the design and biomedical applications of natural polysaccharides. The article is well designed, and the manuscript is well written. The authors are requested to address the following minor comments.

1. The authors should include examples of natural crosslinkers and enzymatic catalysts for polysaccharide hydrogel formation in sub-section 3.1.

2. In Ln 260 sub-section 4.2.1, the author mentioned that "The structure of the hydrogel (polysaccharide-based) is similar to the extracellular matrix." My understanding is that the native extracellular matrix is fibrous in nature. Authors should kindly give examples of polysaccharide-based hydrogels with ECM-like structures.

3. Kindly check Ln 344-345.

4. Why include Science Direct in references 13, 14 and 16?

Reviewer 2 Report

In this review article, the authors have summarized several common polysaccharides for hydrogel synthesis, the synthesis methods of polysaccharide hydrogels, and the functional composite hydrogel dressings in recent years. However, a few issues need to be fixed in the presented review:

1. Please use the term "review," not "paper," throughout the presented manuscript.

2. The introduction must be re-written, keeping in mind the title of the review, "Role of Natural Polysaccharide-based Hydrogel Wound Dressings."

3. Figure 1. represents the mechanism of hydrogel wound dressing, however, it can be beneficial for the reader to see polysaccharide-based hydrogel wound dressing.
4. The authors have only given examples of drug delivery, but it is suggested to show at least one pictorial presentation of all preparation methods of polysaccharide hydrogels.

5. Please provide some insights into the biomedical applications of polysaccharide-based hydrogels.

Reviewer 3 Report

The manuscript presents a very interesting review on the application of polysaccharides for hydrogels formulation in biomedicals. 

The topic correlates to the journal.

The abstract reports a consistent summary of the article findings.

The work has a clear structure.

All sections are required for a complete understanding.

Nevertheless, there are some minor issues that require to be addressed before proceeding with the publication, to enhance the quality and presentation to a broad audience.

An English check would strongly boost the whole manucript.

Check for typos.

A list of possible targeted diseases for the presented hydrogel target would foster the article findings.

References list could benefit additional, broader state-of-the-art sources, with reference the very interesting approach provided over the review (e.g. J. Biomed. Mater. Res. A, 104A (2016) 1668-1679).

The conclusion section would benefit further explanation, e.g. addition of a few sentences recapitulating the whole main focus, the scientific progress and soundness of the review work. It might help discuss over an additional section any possible limitations/future perspectives.

Round 2

Reviewer 2 Report

Although the review has improved, several concerns still need to be carefully looked at and fixed.
The authors have updated the introduction to make it clearer how natural polysaccharide-based wound dressings differ from synthetic hydrogel wound dressings and where natural polysaccharides come from as well as how these dressings affect wound healing. However, the writers failed to cite the updated content.
The introduction as it is now written also lacks a theme or a distinct plot thread.
Hydrogels in Figure 1 have only been replaced with polysaccharide hydrogels by Auhors. Actually, Figure 1 needs to be redrew to better fit the idea of the paper. Figure 1 can be improved by reading several outstanding review papers.
Errors in grammar must be carefully removed.
Additionally, kindly review the quality of the figures included in the updated manuscript.
Therefore, it is advised that these issues be revised before moving on to further considerations.
